# AN ETHICAL MODULE FOR BIAS MITIGATION OF PRE-TRAINED FACE RECOGNITION MODELS

## ABSTRACT

In spite of the high performance and reliability of deep learning algorithms in broad range everyday applications, many investigations tend to show that a lot of models exhibit biases, discriminating some subgroups of the population. This urges the practitioner to develop fair systems whose performances are uniform among individuals. In this work, we introduce a post-processing method designed to mitigate bias of state-of-the-art models. It consists in learning a shallow neural network, called the *Ethical Module*, which transforms the deep embeddings of a pre-trained model to give more representation power to the discriminated subgroups. Its training is supervised by the von Mises-Fisher loss, whose hyperparameters allow to control the space allocated to each subgroup in the latent space. Besides being very simple, the resulting methodology is more stable and faster than most current methods of bias mitigation. In order to illustrate our idea in a concrete use case, we focus here on gender bias in facial recognition and conduct extensive numerical experiments on standard datasets.

## 1 INTRODUCTION

In the past few years, Face Recognition (FR) systems have reached extremely high levels of performance, paving the way to a broader range of applications, where the reliability levels were previously prohibitive to consider automation. This is mainly due to the adoption of deep learning techniques in computer vision since the famous breakthrough of Krizhevsky et al. (2012). The increasing use of deep FR systems has however raised concerns as any technological flaw could have strong societal impact. Besides recent punctual events[1] that received significant media coverage, the academic community has studied bias of FR systems since many years (dating back at least to Phillips et al. (2003) who investigated racial bias of non-deep FR algorithms). Abdurrahim et al. (2018) identify three sources of biases: race (understood as biological attributes such as skin color), age Srinivas et al. (2019) and gender Albiero et al. (2020). The National Institute of Standards and Technology Grother et al. (2019) conducted a thorough analysis of the performances of several FR algorithms in function of these attributes and revealed high disparities. For instance, some of the top state-of-the-art algorithms in absolute performances have more than seven times false acceptances for females than for males. In this paper, we introduce a methodology to mitigate gender bias for FR. Though focusing on a single source of bias has obvious limitations regarding intersectional effects Buolamwini & Gebru (2018), it is a first step to gain insights into the mechanisms at work, before turning to more complex situations. Actually, the method promoted in this paper, much more general than the application considered here, could possibly alleviate many other types of bias. This will be the subject of a future work.

The topic corresponding to the study of different types of bias and to the elaboration of methods to alleviate them is referred to as *fairness* in machine learning, which has received increasing attention in recent years Mehrabi et al. (2019), Caton & Haas (2020), Du et al. (2020). Roughly speaking, achieving fairness means learning an algorithm that does not mistreat some predefined subgroups, while still exhibiting a good predictive performance on the overall population: in general, a trade-off has to be found between fair treatment and pure accuracy[2]. In this regard, one needs to carefully

---

[1]See for instance the study conducted by the American Civil Liberties Union.

[2]this dichotomy somewhat simplifies the problem since an increase in accuracy could also lead to a better treatment of each subgroup of the population.

define what will be the relevant *fairness metric*. From a theoretical viewpoint, several ones have been introduced Castelnovo et al. (2021), Garg et al. (2020) depending on how the concept of equity is understood. In practice, these very refined notions can be inadequate, as they ignore specific use case issues, and one thus needs to adapt them carefully. This is particularly the case in FR, where high security standards cannot be negotiated. One of the contributions of our work is to introduce two new metrics that incorporate the needs for both security and fairness (see section 2.2). Once the metric has been chosen, different strategies can be considered to alleviate bias of algorithms which can be roughly grouped in three categories: pre-, in- and post-processsing methods Caton & Haas (2020), depending on whether the practitioner "fairness" intervention occurs before, during or after the training phase.

In this work, we introduce a novel post-processing method allowing to correct gender bias of FR pre-trained models. It is based on the idea that state-of-the-art models Wang et al. (2018), Deng et al. (2019a), Huang et al. (2020) offer already very good deep embeddings of face images, as witnessed by their high performances on large scaled evaluation datasets (*e.g.* IJB-C Maze et al. (2018)), although they could induce various bias. This indicates that the deep face embeddings obtained by these models are quite relevant and can be considered as initialization points to be enhanced. As a result, our methodology consists in learning a shallow Multi-Layers Perceptron (MLP) in order to transform the deep embeddings of the pre-trained model and balance the representation space used by the discriminated groups. The training of this shallow network is supervised by the von Mises-Fisher loss which is particularly well-suited for our purpose. Indeed, it incorporates an hyperparameter for females (resp. males), whose variation is directly linked with the area they cover in the latent space. To emphasize on both the post-processing aspect and the sought fairness, we call our methodology the *Ethical Module*.

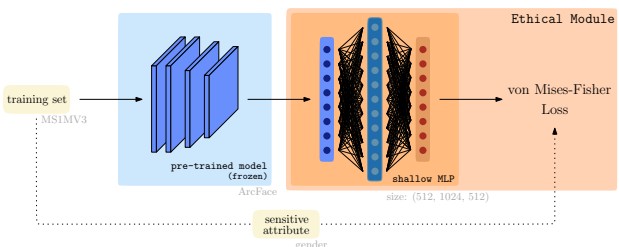

Figure 1: Illustration of the Ethical Module methodology. In gray: our experiment choices.

The *Ethical Module* enjoys several benefits we would like to highlight.

- **Simplicity and interpretability.** Though it is of great simplicity, the method exhibits good performance. In addition, the von Mises-Fisher loss has a nice parametric interpretation which eases the understanding of the debiasing mechanisms at work.

- **Computational complexity and stability.** The training of the shallow MLP is fast (few hours) and avoids the re-training of a full large-scale model, which can be costly and time-consuming. Moreover, the training process is stable and supports the reliability of the method. This is in contrast with adversarial methods whose training losses tend to oscillate. Finally, our method is flexible and we conjecture it could use small database to already debias representation since the shallow MLP has only few parameters.

- **Taking advantage of foundation models.** In the recent survey Bommasani et al. (2021), the authors judiciously point out a change of paradigm in deep learning: very efficient pre-trained models with billions of parameters they call *foundation models* are at our disposal such as BERT Devlin et al. (2018) in NLP or ArcFace Deng et al. (2019a) in FR. Many works rely on these powerful models and fine tune them, inheriting from both their strengths and weaknesses such as their biases. Hence the need to focus on methods to investigate the fairness of foundation models: our method is in line with this approach.

- **Compliance with the fairness through unawareness principle.** If the Ethical Module requires access to the sensitive label during its training phase, the latter only appears in the von Mises-Fisher Loss. As a result, face images are the only argument of the learned embedding function.

**Organization of the paper.** Section 2.1 presents the widely spread usage of FR and its main challenges. It is followed by section 2.2 where we discuss different fairness metrics that arise in FR and introduce two new ones we think are more relevant with regards to operational use cases. In section 3, we present the von Mises-Fisher loss that is used for the training of the Ethical Module and discuss its benefits. Finally, in section 4, we present at length our numerical experiments, which consist in learning an Ethical Module on the ArcFace model, pre-trained on the MS1MV3 dataset (Deng et al. (2019b)). Our results show that, remarkably, some specific choices of hyperparameters provide high performance and low fairness metrics both at the same time.

**Related works.** The correction of bias in FR has been the subject of several recent papers. Liu et al. (2019) and Wang & Deng (2020) use reinforcement learning to learn fair decision rules but despite their mathematical relevance, such methods are computationally prohibitive. Another line of research followed by Yin et al. (2019), Wang et al. (2019a) and Huang et al. (2019) assume that bias come from the unbalanced nature of FR datasets and builds on imbalanced and transfer learning methods. Unfortunately, these methods do dot completely remove bias and it has been recently pointed out that balanced dataset are actually *not enough* to mitigate bias Wang et al. (2019b). Gong et al. (2019), Alasadi et al. (2019) and Dhar et al. (2020; 2021) rely on adversarial methods that can reduce bias but are also known to be unstable and computationally expensive. All of the previously mentioned methods try to learn fair representations. In contrasts, some other works do not affect the latent space but modify the decision rule instead: Terhörst et al. (2020) act on the score function whereas Salvador et al. (2021) rely on calibration methods. Despite encouraging results, these approaches do not solve the source of the problem which is the bias incurred by the embeddings used.

## 2 ON FAIRNESS IN THE CONTEXT OF DEEP FACE RECOGNITION

In this section, we first briefly recall the main principles of deep face recognition and introduce some notations. The interested reader may consult Masi et al. (2018) or Wang & Deng (2018) for a detailed exposition. Then, we present the fairness metrics we adopt and argue of their relevance in our framework.

### 2.1 A QUICK OVERVIEW OF FACE RECOGNITION

A typical FR dataset consists of $N$ images $(\boldsymbol{x}_i)_{1 \leq i \leq N}$ of faces that have been pre-processed and are all of size $(h, w, c)$. It is assumed that there are $K$ identities among the images and we denote by $y_i \in \{1, \ldots K\}$ the identity of $\boldsymbol{x}_i$ for $i = 1, \ldots, N$. The goal of a FR algorithm is to learn a proper $d$-dimensional representation of the face images, by means of a function $f : \mathbb{R}^{h \times w \times c} \rightarrow \mathbb{R}^d$, in order to minimize the intra-identity distances and maximize the inter-identity distances. We denote by $\boldsymbol{z}_i = f(\boldsymbol{x}_i)$ the *face embedding* of $\boldsymbol{x}_i$. Since the advent of deep learning, the function $f$ is usually a deep Convolutional Neural Network (CNN) whose parameters are learned on a large FR dataset.

**Test phase.** There are generally two FR use cases: *identification*, which consists in finding the specific identity of a probe face among several previously enrolled identities, and *verification* (which we focus on throughout this article), which aims at deciding whether two face images correspond to the same identity or not. To do so, the closeness between two embeddings is usually quantified with the cosine similarity measure $s(\boldsymbol{z}_i, \boldsymbol{z}_j) := \boldsymbol{z}_i^\mathsf{T} \boldsymbol{z}_j / (||\boldsymbol{z}_i|| \cdot ||\boldsymbol{z}_j||)$, where $|| \cdot ||$ stands for the usual Euclidean norm (the Euclidean metric $||\boldsymbol{z}_i - \boldsymbol{z}_j||$ is also used in some early works *e.g.* Schroff et al. (2015)). Therefore, an operating point $t \in [-1, 1]$ has to be chosen to classify a pair $(\boldsymbol{z}_i, \boldsymbol{z}_j)$ as *genuine* (same identity) if $s \geq t$ and *impostor* (distinct identities) otherwise.

**Training.** For the training phase only, a fully-connected layer is added on top of the deep embeddings so that the output is a $K$-dimensional vector, predicting the identity of each image within the training set. The full model (CNN + fully-connected layer) is trained as an identity classification task. Until 2018, most of the popular FR loss functions were of the form:

$$\mathcal{L} = -\frac{1}{n} \sum_{i=1}^{n} \log \left( \frac{e^{\kappa \boldsymbol{\mu}_{y_i}^\mathsf{T} \boldsymbol{z}_i}}{\sum_{k=1}^{K} e^{\kappa \boldsymbol{\mu}_k^\mathsf{T} \boldsymbol{z}_i}} \right), \tag{1}$$

where the $\boldsymbol{\mu}_k$'s are the fully-connected layer's parameters, $\kappa > 0$ is the inverse temperature of the softmax function used in brackets and $n$ is the batch size. Early works (Taigman et al. (2014); Sun

et al. (2014)) took $\kappa = 1$ and used a bias term in the fully-connected layer but Wang et al. (2017) showed that the bias term degrades the performance of the model. It was thus quickly discarded in later work. Since the canonical similarity measure at the test stage is the cosine similarity, the decision rule only depends on the angle between two embeddings, whereas it could depend on the norms of $\boldsymbol{\mu}_k$ and $\boldsymbol{z}_i$ during training. This has led Wang et al. (2017) and Hasnat et al. (2017) to add a normalization step during training and take $\boldsymbol{\mu}_k, \boldsymbol{z}_i \in \mathbb{S}^{d-1} := \{z \in \mathbb{R}^d : ||z|| = 1\}$ as well as introducing the re-scaling parameter $\kappa$ in Eq. 1: these ideas significantly improved upon former models and are now widely adopted. Denoting by $\theta_i$ the angle between $\boldsymbol{\mu}_{y_i}$ and $\boldsymbol{z}_i$, the major advance over the loss of Eq. 1 (with normalization of $\boldsymbol{\mu}_k, \boldsymbol{z}_i$) in recent years was to consider large-margin losses which replace $\boldsymbol{\mu}_{y_i}^\intercal \boldsymbol{z}_i = \cos(\theta_i)$ by a function that reduces intra-class angle variation, such as the $\cos(m\theta_i)$ of Liu et al. (2017) or the $\cos(\theta_i) - m$ of Wang et al. (2018). The most efficient choice is $\cos(\theta_i + m)$ and is due to Deng et al. (2019a) who called their model ArcFace, on which we build our methodology.

A fine training should result in the alignment of each embedding $\boldsymbol{z}_i$ with the vector $\boldsymbol{\mu}_{y_i}$. The aim is to bring together embeddings with the same identity. Indeed, during the test phase, the learned algorithm will have to decide whether two face images are related to the same, potentially unseen, individual (one refers to an *open set* framework).

**Evaluation metrics.** Denoting by $\mathcal{G}$ the set of genuine pairs and by $\mathcal{I}$ the set of impostor pairs in a given test set, we introduce the False Acceptance and False Rejection Rates, defined as follows:

$$\mathrm{FAR}(t) := \frac{\#\{(\boldsymbol{z}_i, \boldsymbol{z}_j) \in \mathcal{I} : s(\boldsymbol{z}_i, \boldsymbol{z}_j) \geq t\}}{\#\{(\boldsymbol{z}_i, \boldsymbol{z}_j) \in \mathcal{I}\}}, \quad \mathrm{FRR}(t) := \frac{\#\{(\boldsymbol{z}_i, \boldsymbol{z}_j) \in \mathcal{G} : s(\boldsymbol{z}_i, \boldsymbol{z}_j) < t\}}{\#\{(\boldsymbol{z}_i, \boldsymbol{z}_j) \in \mathcal{G}\}}.$$

These quantities are crucial to evaluate a given algorithm in our context: face recognition is intrinsically linked to biometric applications, where the usual accuracy evaluation metric is not sufficient to assess the quality of a learned decision rule. For instance, security automation in an airport requires a very low FAR while keeping a reasonable FRR to ensure a pleasant user experience. As a result, the most widely used metric consists in first fixing a threshold $t$ so that the FAR is equal to a pre-defined value $\alpha \in [0, 1]$, and then computing the FRR at this threshold. We use the *canonical FR notation* to denote the resulting quantity:

$$\mathrm{FRR}@(\mathrm{FAR} = \alpha) := \mathrm{FRR}(t) \text{ with } t \text{ such that } \mathrm{FAR}(t) = \alpha. \tag{2}$$

The FAR level $\alpha$ determines the operational point of the FR system and corresponds to the security risk one is ready to take. According to the use case, it is typically set to $10^{-i}$ with $i \in \{1, \ldots, 6\}$.

## 2.2 Fairness Metrics

While the FRR@FAR metric is the standard choice for measuring the performance of a FR algorithm, it does not take into account its variability among different subgroups of the population. In order to assess and correct for potential discriminatory biases, the practitioner must rely on suitable fairness metrics. The first step is to define a finite set $\mathcal{A}$ of *sensitive attributes* against which we wish to assess fairness. Since we focus here on gender bias, we take $\mathcal{A} = \{0, 1\}$ where 0 stands for "male" and 1 for "female". We extend the notions of FAR and FRR within the male (resp. female) subgroups, considering same-gender pairs only, and denote by $\mathrm{FAR}_0, \mathrm{FAR}_1$ and $\mathrm{FRR}_0, \mathrm{FRR}_1$ the resulting quantities.

Before specifying our choice for the fairness metric used here, let us review some existing ones Cavazos et al. (2020) that derive from fairness in the context of binary classification (here, one classifies pairs in two groups: genuines or impostors). The *Demographic Parity* criterion requires the prediction to be independent of the sensitive attribute, which amounts to equalizing the likelihood of being genuine conditional to $a = 0$ and $a = 1$. Besides heavily depending on the number and quality of impostors and genuines pairs among subgroups, this criterion does not take into account the FARs and FRRs, which are instrumental in FR as previously mentioned. An attempt to incorporate those criteria could be to compare the intra-group performances: $\mathrm{FRR}_0@(\mathrm{FAR}_0 = \alpha)$ v.s. $\mathrm{FRR}_1@(\mathrm{FAR}_1 = \alpha)$. However, the operational points $t_0$ and $t_1$ satisfying $\mathrm{FAR}_0(t_0) = \alpha$ and $\mathrm{FAR}_1(t_1) = \alpha$ generically differ as pointed out by Krishnapriya et al. (2020). To fairly asses the equity of an algorithm, one needs to compare intra-groups FARs and FRRs at a same threshold. Two such criteria exist in the fairness literature: the *Equal Opportunity* fairness criterion which requires $\mathrm{FRR}_0(t) = \mathrm{FRR}_1(t)$ and the *Equalized Odds* criterion which additionally requires $\mathrm{FAR}_0(t) =$

$\text{FAR}_1(t)$. Nevertheless, working at an arbitrary threshold does not really make sense since, as previously mentioned, FR systems typically choose an operational point achieving a predefined FAR level so as to limit security breaches. This is why most current papers consider a fixed operational point $t$ such that the population False Acceptance Rate equals a fixed value $\alpha$. For instance, Dhar et al. (2020) computes

$$|\text{FRR}_1(t) - \text{FRR}_0(t)| \quad \text{with } \text{FAR}(t) = \alpha. \tag{3}$$

However, we think the choice of a threshold achieving a global FAR is not entirely relevant for it depends on the relative proportions of females and males of the considered dataset together with the relative proportion of intra-groups impostors and genuines. For instance, at fixed images quality, if females represent a small proportion of the dataset, the threshold $t$ of Eq. 3 is close to the male threshold $t_0$ satisfying $\text{FAR}_0(t_0) = \alpha$ and away from the female threshold $t_1$ satisfying $\text{FAR}_1(t_1) = \alpha$. Such a variability among datasets could lead to incorrect conclusions.

In this paper, we go one step further and work at a threshold achieving $\max_a \text{FAR}_a = \alpha$ instead of $\text{FAR} = \alpha$. This alleviates the previously proportions dependence. Besides, this allows to monitor the risk one is willing to take among each subgroup: for a pre-definite rate $\alpha$ deemed acceptable, one typically would like to compare the performance among subgroups for a threshold where *each* subgroup satisfies $\text{FAR}_a \leq \alpha$. Our two resulting metrics are thus:

$$\text{BFRR}(\alpha) := \frac{\max_{a \in \{0,1\}} \text{FRR}_a(t)}{\min_{a \in \{0,1\}} \text{FRR}_a(t)} \quad \text{with } t \text{ such that} \quad \max_{a \in \{0,1\}} \text{FAR}_a(t) = \alpha \tag{4}$$

and

$$\text{BFAR}(\alpha) := \frac{\max_{a \in \{0,1\}} \text{FAR}_a(t)}{\min_{a \in \{0,1\}} \text{FAR}_a(t)} \quad \text{with } t \text{ such that} \quad \max_{a \in \{0,1\}} \text{FAR}_a(t) = \alpha. \tag{5}$$

One can read the above acronyms "Bias in FRR/FAR". In addition to being more security demanding than previous metrics, BFRR and BFAR are more amenable to interpretation: the ratios of FRRs or FARs corresponds to the number of times the algorithm makes more mistakes on the discriminated subgroup. Those metrics generalizes well for more than 2 distinct values of the sensitive attribute.

## 3  THE VON MISES-FISHER LOSS FOR BIAS MITIGATION

We now turn to a detailed description of the von Mises-Fisher (vMF in abbreviated form) loss which supervises the training of our Ethical Module's MLP. It stems from the vMF probability measure on the hypersphere and was first introduced in Hasnat et al. (2017) as a new powerful way to discriminate different identities, before the advent of large-margin losses.

**The von Mises-Fisher distribution.** The vMF distribution in dimension $d$ with mean direction $\boldsymbol{\mu} \in \mathbb{S}^{d-1}$ and concentration parameter $\kappa > 0$ is a probability measure defined on the hypersphere $\mathbb{S}^{d-1}$ by the following density:

$$V_d(\boldsymbol{x}; \boldsymbol{\mu}, \kappa) := C_d(\kappa) e^{\kappa \boldsymbol{\mu}^\intercal \boldsymbol{x}} \quad \text{with } C_d(\kappa) = \frac{\kappa^{\frac{d}{2}-1}}{(2\pi)^{\frac{d}{2}} I_{\frac{d}{2}-1}(\kappa)},$$

where $I_\nu$ stands for the modified Bessel function of the first kind at order $\nu$, whose logarithm can be computed with high precision (Kim (2021)). Figure 2 illustrates the influence of the concentration parameter $\kappa$ on the vMF distribution.

The vMF distribution corresponds to a Gaussian distribution in dimension $d$ with mean $\boldsymbol{\mu}$ and variance-covariance matrix $(1/\kappa)\boldsymbol{I_d}$, conditioned to live on the hypersphere. It makes it a very natural *directional law*: for instance, it has maximal entropy among all probability measures on the hypersphere with fixed variance.

**A Fair von Mises-Fisher loss.** The general form of the vMF Loss on the face embeddings is

$$\mathcal{L}_{\text{vMF}} = -\frac{1}{n} \sum_{i=1}^{n} \log \left( \frac{C_d(\kappa_{y_i}) e^{\kappa_{y_i} \boldsymbol{\mu}_{y_i}^\intercal \boldsymbol{z}_i}}{\sum_{k=1}^{K} C_d(\kappa_k) e^{\kappa_k \boldsymbol{\mu}_k^\intercal \boldsymbol{z}_i}} \right), \tag{6}$$

where a scalar $\kappa_k$ is affected to each identity $k \in \{1, \ldots, K\}$. Notice that there are two ways of minimizing $\mathcal{L}_{\text{vMF}}$: either by aligning ground-truth $\boldsymbol{\mu}_{y_i}$ with associated normalized face embeddings

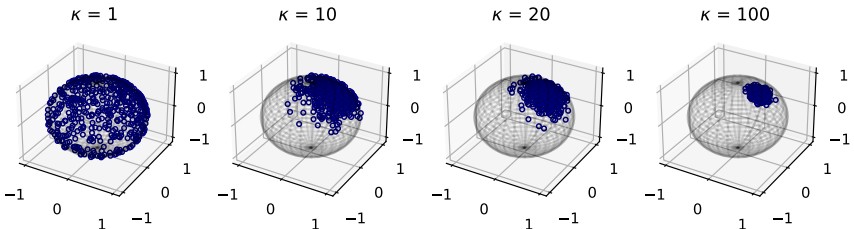

Figure 2: 500 samples from the vMF in dimension 3 with parameters $\mu = [0.5, 0, \sqrt{0.75}]$ and $\kappa$.

$\boldsymbol{x}_i$ or by pushing back wrong $\boldsymbol{\mu}_k$ (with $k \neq y_i$) from $\boldsymbol{x}_i$. The vMF Loss was first introduced in the context of FR by Hasnat et al. (2017), who took a unique value $\kappa$. In that case, the vMF loss reduces to the classical loss $\mathcal{L}$ of Eq. 1 (when $\boldsymbol{z}_i$ and $\boldsymbol{\mu}_i$ are normalized). This makes $\mathcal{L}_{\text{vMF}}$ a natural generalization of cosine similarity-based losses which are at the basis of refined state-of-the-art models that also incorporate margins.

Building on $\mathcal{L}_{\text{vMF}}$, we consider gender-based hyperparameters. If $a_k \in \{0, 1\}$ denotes the gender of identity $k \in \{1, \ldots, K\}$, the fair version of the vMF-loss we consider to train the Ethical Module is:

$$\mathcal{L}_{\text{FvMF}} = -\frac{1}{n} \sum_{i=1}^{n} \log \left( \frac{C_d(\kappa_{a_{y_i}}) e^{\kappa_{a_{y_i}} \boldsymbol{\mu}_{y_i}^{\mathsf{T}} \boldsymbol{z}_i}}{\sum_{k=1}^{K} C_d(\kappa_{a_k}) e^{\kappa_{a_k} \boldsymbol{\mu}_k^{\mathsf{T}} \boldsymbol{z}_i}} \right). \tag{7}$$

It depends on two hyper-parameters $\kappa_0, \kappa_1 > 0$, respectively corresponding to the concentration parameters of male and female. By tuning $\kappa_0$ and $\kappa_1$, one can modify the surfaces covered by the male / female and therefore plays on their representative powers.

**A maximum likelihood interpretation of the von Mises-Fisher loss.** It is common knowledge that, besides its high performances in many contexts, the Cross-Entropy (CE) loss is also attractive because of its interpretation in terms of maximum likelihood estimation. Indeed, minimizing the CE amounts to maximizing the likelihood with respects to the network parameters.

Interestingly, the vMF loss also enjoys this property if one places a proper model on the face embeddings. Let us describe this model, which is a mixture of vMF distributions. Recall that $K \geq 1$ is the number of identities in the dataset. The model consists in assuming that the dataset consists of i.i.d. realizations of a probability law with density:

$$g_d \left( \boldsymbol{x}; \{(\pi_k, \boldsymbol{\mu}_k, \kappa_k)\}_{1 \leq k \leq K} \right) := \sum_{k=1}^{K} \pi_k V_d(\boldsymbol{x}; \boldsymbol{\mu}_k, \kappa_k).$$

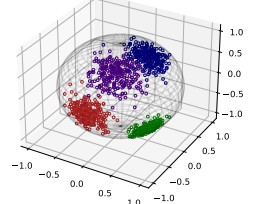

Let us discuss its parameters $\boldsymbol{\Theta}_{\text{vMF}} := \{(\pi_i, \boldsymbol{\mu}_i, \kappa_i)\}_{1 \leq i \leq K}$. Each identity $k \in \{1, \ldots, K\}$ is represented by a vMF distribution with mean $\boldsymbol{\mu}_k$ and concentration $\kappa_k$. In this vision, the mean $\boldsymbol{\mu}_k$ is an abstraction of the identity and $\kappa_k$ takes into account its variability (caused by lightening conditions, aging...). We also denote by $\pi_k$ the proportion of identity $k$ in the overall population. Figure 3 illustrates this vMF mixture in the case $K = 4$ (the detail of the

Figure 3: Illustration of a vMF mixture model.

parameters is given in the supplementary material). Under the vMF mixture model assumption, the probability $p_{ij}$ that a face embedding $\boldsymbol{z}_i$ belongs to identity $k$ is given by

$$p_{ij} = \frac{V_d(\boldsymbol{z}_i | \boldsymbol{\mu}_j, \kappa_j)}{\sum_{k=1}^{K} V_d(\boldsymbol{z}_i | \boldsymbol{\mu}_k, \kappa_k)} = \frac{\pi_j C_d(\kappa_j) e^{\kappa_j \boldsymbol{\mu}_j^T \boldsymbol{z}_i}}{\sum_{k=1}^{K} \pi_k C_d(\kappa_k) e^{\kappa_k \boldsymbol{\mu}_k^T \boldsymbol{z}_i}}.$$

Therefore, the negative log-likelihood of the model is given by

$$\text{NLL}(\boldsymbol{\Theta}, \boldsymbol{\Theta}_{\text{vMF}}) := -\frac{1}{N} \sum_{i=1}^{N} \log \left[ \frac{\pi_{y_i} C_d(\kappa_{y_i}) \, e^{\kappa_{y_i} \, \boldsymbol{\mu}_{y_i}^T \boldsymbol{z}_i}}{\sum_{k=1}^{K} \pi_k C_d(\kappa_k) \, e^{\kappa_k \, \boldsymbol{\mu}_k^T \boldsymbol{z}_i}} \right].$$

At least at a heuristical level, both playing on the parameters $\pi_k$s and $\kappa_k$s affects the surfaces covered by the different classes. It is thus natural to restrict the scope to the case where the $\pi_k$s are all equal and only adjust the concentration parameters. In that case, the above NLL is in fact the vMF Loss $\mathcal{L}_{vMF}$ previously introduced.

## 4 Numerical Experiments

**Pre-trained model.** We use the trained model ArcFace[3] whose CNN architecture is a ResNet100 (Han et al. (2017)). As emphasized before, it achieves state-of-the-art performances in FR. It has been trained on the MS1M-RetinaFace dataset (also called MS1MV3), introduced by Deng et al. (2019b) in the ICCV 2019 Lightweight Face Recognition Challenge. MS1MV3 is a cleaned version of the MS-Celeb1M dataset (Guo et al. (2016)); all its face images have been pre-processed by the Retina-Face detector of Deng et al. (2019c) and are of size $112 \times 112$ pixels. It contains 5.1M images of 93k identities. We also consider other pre-trained models[4] (AdaCos Zhang et al. (2019), CosFace Wang et al. (2018), CurricularFace Huang et al. (2020)) whose backbone is a Mobile-FaceNet Chen et al. (2018) and trained on the MS-Celeb-1M-v1c-r dataset[5]. This dataset is another cleaned version of the MS-Celeb1M dataset and it contains 3.28M images of 73k identities. The images are also pre-processed by the Retina-Face detector and are of size $112 \times 112$ pixels.

**Gender labels.** For a fair comparison, we train our Ethical Module on the training set used to train the pre-trained models (MS1MV3 for ArcFace, MS-Celeb-1M-v1c-r for the models with Mobile-FaceNet backbone). However, ground-truth gender labels for MS1MV3/MS-Celeb-1M-v1c-r are not available. As the training of our Ethical Module needs the gender label of each face image within the training set, we use a private gender classifier to get those gender labels. Current gender classifiers achieve around 95% prediction accuracy on standard evaluation datasets and are widely used in FR to get gender annotations (Acien et al. (2018); Gong et al. (2020)). Since some images from the same identity might be assigned different gender predictions, it is common practice to use a majority vote to decide the correct gender for each identity. We follow Albiero et al. (2020) and only keep in our training sets the identities for which at least 75% of the same-identity face images are assigned the same gender. Doing so, we discard 25k images and 835 identities for MS1MV3, 10k images and 500 identities for MS-Celeb-1M-v1c-r.

**Ethical Module.** The face embeddings output by the pre-trained models are of dimension 512. Thus, the MLP within our Ethical Module has an input layer of 512 units. To emphasize the fact that our gender bias mitigation solution is much less costly than current solutions such as Wang & Deng (2020) and Dhar et al. (2020) , in terms of both training time and computation power (see supplementary material A.1), we choose a shallow MLP of size (512, 1024, 512), the output dimension being the same than for Arcface. This MLP is trained with the fair version $\mathcal{L}_{\text{FvMF}}$ of the vMF loss introduced in Eq. 7. For each experiment, we train the Ethical Module during 50 epoch with the Adam optimizer (Kingma & Ba (2014)). The batch size is set to 1024 and the learning rate to 0.01. The training is efficient as we first compute the face embeddings of the pre-trained models (on MS1MV3 for ArcFace, on MS-Celeb-1M-v1c-r for the models with MobileFaceNet backbone), store them, and then train a shallow MLP on those embeddings. Using one single GPU (NVIDIA RTX 3090), the computation of the embeddings takes 4 hours and each training takes 8 hours.

**Reproducibility.** We plan to release the code used to conduct our experiments.

### 4.1 Grid-Search on IJB-C and fairness evaluation

In order to select relevant pairs of gender-hyperparameters $(\kappa_0, \kappa_1)$, we perform a grid-search and keep track of the canonical performance metric $\text{FRR@}(\text{FAR} = 10^{-3})$ together with our two fairness

---

[3]https://github.com/deepinsight/insightface/tree/master/recognition/arcface_torch.
[4]https://github.com/JDAI-CV/FaceX-Zoo/blob/main/training_mode/README.md.
[5]See footnote 4.

metrics $\text{BFRR}(10^{-3})$ and $\text{BFAR}(10^{-3})$ introduced in Eq. 4 and 5. To obtain reliable results, we need to compute the latter metrics on a sufficiently large dataset containing gender labels. We choose IJB-C (Maze et al. (2018)), which contains about 3,5k identities for a total number of about 31k images and 117k unconstrained video frames. The 1:1 verification protocol[6] is performed on 19k genuine pairs and 15M impostor pairs.

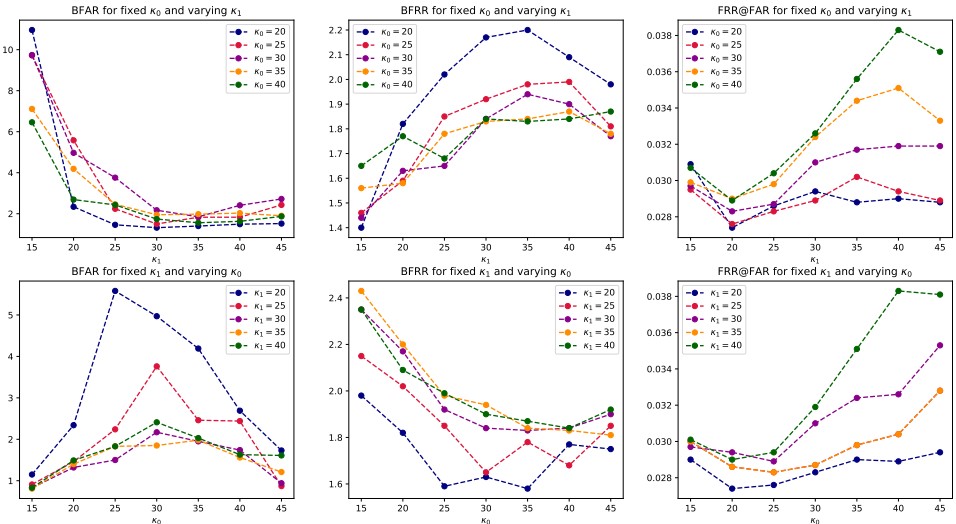

Figure 4: Ablation study of the considered metrics when one of the two hyperparameters is fixed. The FAR level defining the threshold $t$ is set to $10^{-3}$; the pre-trained model is ArcFace with a ResNet100 backbone.

A heatmap representation of the grid-search is provided in the supplementary material A.2. Several interesting trends emerge from the ablation study of Figure 4, suggesting an underlying regularity of the model with respect to the hyperparameters space. More precisely:

- when $\kappa_0$ is fixed and $\kappa_1$ increases, BFAR tends to decrease, BFRR first increases and then decreases and FRR@FAR tends to increase,

- when $\kappa_1$ is fixed and $\kappa_0$ increases, BFAR first increases and then decreases, BFRR tends to decrease and and FRR@FAR increases.

A strength of our approach is that it lends itself well to a geometric interpretation. The latter allows to derive two heuristic views on the problem, whose details are provided in the supplementary material A.3. In a nutshell, we are able to *(i)* identify a pair of hyperparameters with *a priori* good fairness properties, without relying on a grid-search study, and *(ii)* provide explanations for the trends of Figure 4. A very promising route for future work would be to investigate a two-step strategy for finding the best hyperparameters by selecting an initial point $(\kappa_0, \kappa_1)$ with *(i)* and then navigating the hyperparameter space with *(ii)*.

Many $(\kappa_0, \kappa_1)$ pairs could be considered as relevant and instead of defining an objective criterion, we select three of them in order to illustrate the trade-off one needs to make between fairness metrics and pure performance. The point $(\kappa_0 = 25, \kappa_1 = 20)$ is quite relevant regarding BFRR and performance (FRR@FAR), but is not adapted for BFAR. On the other hand, the point $(15, 20)$ is good when it comes to BFAR and performance. Finally, the point $(45, 30)$ is interesting when considering BFRR and BFAR. In Table 1, we summarize the different metrics evaluated for these three points on the LFW dataset (Huang et al. (2008), see A.4) and compare them with the ArcFace pre-trained baseline, at two FAR levels. We also conduct these experiments on different pre-trained models with MobileFaceNet backbone. Additional results (using the IJB-C dataset for evaluation, considering several types of ResNet backbones for ArcFace) can be found in the supplementary material A.5.

---

[6]https://github.com/deepinsight/insightface/tree/master/recognition/_evaluation_/ijb.

| | FAR level: | $10^{-4}$ | | | $10^{-3}$ | | |
|---|---|---|---|---|---|---|---|
| | model | FRR@FAR (%) | BFRR | BFAR | FRR@FAR (%) | BFRR | BFAR |
| ArcFace | original | **0.063** | 10.76 | 3.98 | **0.052** | 2.23 | 1.81 |
| | (15,20) | 0.119 | 12.73 | **1.72** | 0.067 | 8.43 | **1.04** |
| | (25,20) | 0.076 | **5.35** | 29.33 | 0.052 | **1.94** | 3.96 |
| | (45,30) | 0.129 | 13.47 | 2.99 | 0.067 | 6.02 | 1.24 |
| AdaCos | original | **2.97** | 3.64 | 3.84 | 0.98 | 5.29 | 2.23 |
| | (15,20) | 4.56 | 4.42 | **1.41** | 1.33 | 6.34 | **1.01** |
| | (25,20) | 3.12 | **2.71** | 8.37 | **0.91** | 4.23 | 3.71 |
| | (45,30) | 4.05 | 4.51 | 1.57 | 1.26 | 7.28 | 1.08 |
| CosFace | original | **1.73** | 5.89 | 2.51 | **0.58** | 8.18 | 1.74 |
| | (15,20) | 3.69 | 5.76 | **1.13** | 1.05 | 8.41 | **1.02** |
| | (25,20) | 2.41 | **3.03** | 9.66 | 0.67 | **5.09** | 4.75 |
| | (45,30) | 2.60 | 4.30 | 3.69 | 0.82 | 6.81 | 1.87 |
| Curricular | original | **2.52** | 3.67 | 2.92 | **0.81** | 4.88 | 1.91 |
| | (15,20) | 3.86 | 5.26 | **1.16** | 1.17 | 6.35 | **1.10** |
| | (25,20) | 2.82 | **2.58** | 9.10 | 0.82 | **3.89** | 4.28 |
| | (45,30) | 3.61 | 3.40 | 2.30 | 1.02 | 5.63 | 1.27 |

Table 1: Evaluation on LFW for ArcFace with ResNet100 backbone and different pre-trained models (AdaCos, CosFace, CurricularFace) with MobileFaceNet backbone. By "original" we mean no Ethical Module is added to the pre-trained model. The tuples correspond to the choices of $\kappa_0$ (first argument) and $\kappa_1$ (second argument). FRR@FAR is expressed as a percentage (%).

## 4.2 Verification evaluation on IJB-B

We finally investigate the FRR@FAR metric (Table 2) of the three selected points $(\kappa_0, \kappa_1)$ on IJB-B (Whitelam et al. (2017)). In the verification setting, this dataset contains 10k genuine pairs and 8M impostor pairs. Notice that we do not lose too much in performance with respect to the original model.

| Methods (%) | $10^{-4}$ | $10^{-3}$ |
|---|---|---|
| original | **5.38** | **3.78** |
| (15, 20) | 6.79 | 4.11 |
| (25, 20) | 6.00 | 3.84 |
| (45, 30) | 7.03 | 4.81 |

Table 2: FRR@(FAR = $\alpha$) on IJB-B for ArcFace with ResNet100 backbone, for $\alpha = 10^{-i}$, $i = 3, 4$.

## 5 Conclusion

In this paper, we introduce a novel method, the *Ethical Module*, to mitigate gender bias of Face Recognition state-of-the-art models. It consists in learning a shallow MLP on top of a frozen pre-trained model, so as to correct the biases that could exist in the embedding space. To achieve fairness, we rely on a fair version of the von Mises-Fisher loss that incorporates an hyperparameter per gender, the variation of which allowing to monitor the space covered by males and females in the latent space. Measuring the fairness of Face Recognition systems is a very challenging task and we introduce two new metrics that both respond to the need for security and equity.

Besides being very simple, the resulting methodology is more stable and faster than most current methods of bias mitigation. It both leverages the strong accuracy of pre-trained models while correcting their bias. We illustrate the soundness of our methodology on several pre-trained models, and strongly believe it could also be used to alleviate other types of bias. Our work opens several lines of research: for instance, it would be interesting to extend our ideas to the context of multiclass sensitive attributes and of continuous sensitive attributes such as age. Another idea would be to somehow incorporate our fairness criteria during the training of the Ethical Module. Finally, we think that incorporating large-margin constraints into the loss used to train the Ethical Module would be a promising attempt to go beyond the trade-off between fairness and performance.

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
