# OpenReview forum: "Learning an Ethical Module for Bias Mitigation of pre-trained Models"
_ICLR.cc/2022/Conference — ICLR 2022 Submitted_

### Official Review · Reviewer_u2Cg · 2021-10-30

**Correctness:** 4
**Technical Novelty And Significance:** 4
**Empirical Novelty And Significance:** 4
**Recommendation:** 5
**Confidence:** 4

**Main Review:**

**Advantages**:
1. It is well written, and I enjoyed reading it.
2. The proposed post-processing module is simple, yet effective.
3. Two new metrics are proposed to help evaluate the fairness of Face Recognition systems.


**I have some minor concerns and questions**:

1. There is no comparing baselines. It is thus difficult to evaluate the relative effectiveness of the proposed model.
2. Experiments section only provides quantitative evaluation results. There is no analysis of what has been learned by the Ethical Module. It would be great if the authors could provide some explainability analysis, to provide insights into the transformation that has been conducted by the Ethical Module.
3. Only gender bias is considered. Could the proposed module be adapted to address other categories of bias, such as race and age bias? In addition, the authors mentioned that the current gender classifiers are used to generate proxy gender labels. If the race bias is considered, whether current race classifiers could be used to generate proxy race labels?
4. For the pre-trained model, only Arcface is considered. It would be interesting to know the mitigation performance for other pretrained models.


=========

I did not see the authors' response. Also considering other reviewers' comments, I have adjusted my rating to 5.


**Summary Of The Paper:**

This paper proposes a post-processing mitigation method, termed as the Ethical Module, to mitigate gender bias of Face Recognition models. The ethical module is a three-layer shallow MLP built on top of a frozen pretrained model, with the goal to correct the biases that could exist in the embedding space of the pretrained model. The module is trained using the proposed von Mises-Fisher loss. Despite being very simple, experimental results on benchmark face recognition datasets have validated the effectiveness of the proposed post-processing mitigation module.


**Summary Of The Review:**

The proposed ethical module and the way to train it is novel. Besides the paper is well written. The experimental evaluation part could be strengthened, to better validate the effectiveness of the proposed method.

---

### Official Review · Reviewer_k7gc · 2021-11-02

**Correctness:** 3
**Technical Novelty And Significance:** 2
**Empirical Novelty And Significance:** 2
**Recommendation:** 5
**Confidence:** 4

**Main Review:**


Strengths:

- The paper proposes an effective post-processing ethical module, which helps to improve the fairness of the pre-trained face recognition models.
- The paper discusses how to measure fairness in the image verification tasks and suggests ratio-based metrics with an additional constraint. The fairness metrics seem to be inspired by Equalized Odds, which is one of the promising group fairness definitions.

Weaknesses:

<Algorithm>

   - Several design choices are unclear. For example, is the von Mises-Fisher loss (vMF loss) used because it has suitable characteristics for combining fairness features into the face recognition framework? Or, was it chosen because vMF is recently-used in the face recognition field? Such details should be added.
   - Also, it is unclear how the final vML loss and the fairness metrics (BFRR, BFAR) are linked. It seems that the current algorithm just adjusts the influence of each group in the loss function through a grid search on hyperparameters. This approach seems a bit naive. Moreover, the effects of the hyperparameters $\kappa_0$ and $\kappa_1$ are not clear. Thus, it would be better to explain why this algorithm is suitable for achieving the fairness metrics.

<Experiments>

- The comparison with existing fairness algorithms is missing. Although the paper mentions several previous algorithms (Wang & Deng, 2020, Dhar et al., 2020), the experiments compared neither fairness nor training efficiency. Also, the paper claims that the proposed algorithm is faster and stable than the previous approaches, but these arguments lack justification.
- It is less convincing that the proposed algorithm would work well in other scenarios. The current experiments only use one pre-trained model (ArcFace), and the fairness performances (i.e., BFAR and BFRR) are mainly tested on the IJB-C dataset. Thus, it would be better if the proposed algorithm is tested on other scenarios as well.
- The fairness results in Section 4.1 are hard to understand. For example, in Figure 4(b), is not the fairness value (FAR1/FAR0) decreased when $\kappa_1$ increased (i.e., opposite from the page 8 explanation)? Also, all the contents in Figure 4 are too small to recognize easily, and the point (20, 26) seems not to appear in Figure 4. The interval of the ticks is five, then where does the number 26 come from? Moreover, it is hard to compare the fairness performances in Figures 4(a), 4(b), and 4(c), as all color bars' scales are different. It would be much better to clarify the descriptions.

-------------------------------------------
After reading the authors' response:

Thanks for your answers and more works on the experiments. I have raised my score to 5 since the experimental results became more convincing and clear. However, I still have some concerns about Sections 3 & 4. I understood that the vMF loss is an appropriate loss function for face recognition, and there is some link between the groups in the data and the vMF loss. However, the fairness metrics that the paper suggested are only empirically connected to the loss design. Thus, the paper needs a more precise explanation of why the simple modification of the vMF loss is enough to improve the target fairness (or types of bias). There may be other choices like adding unfairness penalty terms that can directly minimize fairness loss (which might be related to the brief discussion in the Conclusion section). I am not yet convinced why the current loss design is the most suitable way to improve fairness in face recognition. Based on the above reasons, I decided to raise my score to 5.

**Summary Of The Paper:**

This paper proposes a post-processing approach for fair face recognition, especially the verification tasks (i.e., predicting whether two images correspond to the same group or not). The key idea is to add a shallow MLP on the pre-trained embedding models such as ArcFace, and this new MLP is trained using von Mises-Fisher loss. The paper discusses that fairness in face recognition techniques should be evaluated by comparing two groups’ false rates while preserving a security constraint. The algorithm tries to find good hyperparameters (i.e., the concentration parameters in the loss) for high fairness by grid search.

**Summary Of The Review:**

Although the motivation and the target problem are promising, this paper has a large room for improvement (as explained in the main review). Thus, I believe that this paper is currently not good enough to accept.

---

### Official Review · Reviewer_7eUe · 2021-11-02

**Correctness:** 3
**Technical Novelty And Significance:** 3
**Empirical Novelty And Significance:** 3
**Recommendation:** 5
**Confidence:** 3

**Main Review:**

I find the adaptation of face recognition metrics to the fairness
domain interesting. My main concerns are related to the evaluation of
the model. If I understood correctly, the few values for K0 and K1 are
chosen through grid search. The evaluation presents results for three
such combination of K0,K1. I'm guessing the results are for a test
split of the datasets. I'm curious to understand, if the evaluation
would involve a "deployment scenario", in which a dev/eval split set
is used to choose a particular combination for K0, K1, how would the
results look for the test set. The reason I'm asking this is that the
results are not improved overall for all combinations of K0,K1 and it
is not clear what would happen in a realistic setting.

If I understand correctly, the results obtained for IJB-B dataset
seemed to be inferior to the ones of the original model (Table
3). There is no comment on these results in the paper and I wonder if
there is any intuition why these results are different.


**Summary Of The Paper:**

This paper proposes introducing a shallow network used as a
post-processing stage for face recognition models. The newly
introduced module operates on the encodings generated by the main
module and its main goal is to reduce the bias of the original model
as measured by fairness metrics. The fairness metrics used in this
paper are inspired by equalized odds and adapted to the face
recognition scenarios. Instead of using differences, the metrics
introduced employ the ratios of FRRs and FARs (False Rejection and
False Acceptance Rates, similar to false negatives and false positive
rates). The new module is referred to as "the ethical module". The
ethical module is trained using a loss based on the Von Mises-Fisher
distribution.

The ethical model is evaluated in conjunction with Arcface, which is a
state of the art model for face recognition. The ethical model is used
to mitigate gender bias. Gender labels are generated by an in-house
classifier that assigns gender to each image.

The performance of the model is also verified on several image datasets.



**Summary Of The Review:**

The adaptation of fairness metrics to the face recognition setting is
interesting. The evaluation of the method proposed has some gaps that
lowers the quality of the paper.

---

### Official Review · Reviewer_vzpf · 2021-11-08

**Correctness:** 3
**Technical Novelty And Significance:** 4
**Empirical Novelty And Significance:** 4
**Recommendation:** 5
**Confidence:** 3

**Main Review:**

Strengths
- The introduced idea to use the von Mises-Fisher loss as the training objective to effectively control the space allocated to each protected subgroup in latent space is insightful and useful for the ML ethics community. Including the maximum likelihood interpretation was also a good choice as it also helps to illustrate a different perspective of thinking about it.
- The motivation of BFRR and BFAR is well explained by situating it relative to other fairness metrics

Weaknesses
- The citation style does not follow ICLR conventions
- Glaringly missing analysis in Section 4.2
- Lack of connection between BFAR and BFRR (see below)
- Lack of comparison to other post-processing methods (see below)
- The results shown in Section 4.1 do not seem to be clearly analysed (see below)

Questions
- Why exactly is a balanced dataset not enough to mitigate bias? It is good to explain in related work to more strongly motivate your approach of post-processing.
- Under BFRR and BFAR, we do not compare across multiple sensitive attributes, only the worst performing class and the best performing class. Why is this appropriate in your view?
- Under Section 4, how long did it take to train the MLP? On what hardware? It would be good to give such figures if you are claiming it is a fast post-processing method.
- Table 1 seems misleading to compare because FRR@FAR is a %, but BFRR and BFAR are ratios. It might be helpful to remind the reader in the caption
- In Section 4.1, I am not able to see the relationship between increasing $\kappa_1$ and $FRR_1 / FRR_0$ or $FAR_1 / FAR_0$ because the trend is different at each fixed value of $\kappa_0$. Similarly for $\kappa_0$ at each fixed value of $\kappa_1$. Could you help me understand this more clearly? Also, can you provide an intuitive explanation for the role of $\kappa_0$ and $\kappa_1$ with respect to the ratios?
- Can the BFAR and BFRR discussion be applied to Ethical Module other than a grid search (which suggests a weak link if this is the only use)? How so?
- I also find it surprising that the authors did not compare with other post-processing methods in the experiments. How would you compare your approach to others, given that experiments were not run?

Typo
- Right above section 4: NLL not "NNL is in fact the..."


**Summary Of The Paper:**

The paper introduces a post-processing method they call an Ethical Module, which is composed of a shallow multi-layer perceptron trained with a von Mises-Fisher loss. The authors also introduce two new fairness metrics BFRR and BFAR, which are more security demanding and  interpretable.

The author's claimed contributions are the following:
- 2 fairness metrics: BFRR, BFAR
- The Ethical Module


**Summary Of The Review:**

Overall, I think this paper is marginally below the acceptance threshold. I appreciate the novel approach of the von Mises-Fisher loss for post-processing, and the discussion surrounding how to think about it. My concern is in regard to a more intuitive explanation of the behaviour of the hyperparameters governing the von Mises-Fisher loss, and missing analysis with regard to some experiments or comparisons. Without them, the paper in the current state is difficult to accept even though the idea is technically interesting.

---

### Decision · Program_Chairs · 2022-01-20

**Decision:**

Reject

**Comment:**

The paper proposes a novel post-processing method technique that can mitigate the model bias, called the Ethical Module. It transforms the deep embeddings of a given model to give more representation power to the disadvantaged subgroups.

The idea of ​​resolving discrimination against a specific group through effective post processing is promising, and proposing new metrics for fairness is also a very important and relevant issue.

However, the connection between the technique proposed in this paper and the newly proposed fairness metric is not clear, so the focus of the paper is somewhat lowered. Moreover, several design choices are somewhat unclear and ad-hoc. In particular, although there was a lot of improvement through the rebuttal period, it is difficult to verify the superiority of the proposed method via the experiments in the paper; Direct comparisons with existing methods for fairness is essential, and it seems necessary to consider a hyperparameter selection strategy that can be taken in a practical scenario rather than simply choosing the best performing hyperparameter for the test set.